# Prosody-Based Measures for Automatic Severity Assessment of Dysarthric Speech

**Abner Hernandez** [1] 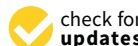**, Sunhee Kim** [2] **and Minhwa Chung** [1,*]

1   Department of Linguistics, Seoul National University, Seoul 08826, Korea; abner@snu.ac.kr
2   Department of French Language Education, Seoul National University, Seoul 08826, Korea; sunhkim@snu.ac.kr
*   Correspondence: mchung@snu.ac.kr; Tel.: +82-2-880-9195

**Abstract:** One of the first cues for many neurological disorders are impairments in speech. The traditional method of diagnosing speech disorders such as dysarthria involves a perceptual evaluation from a trained speech therapist. However, this approach is known to be difficult to use for assessing speech impairments due to the subjective nature of the task. As prosodic impairments are one of the earliest cues of dysarthria, the current study presents an automatic method of assessing dysarthria in a range of severity levels using prosody-based measures. We extract prosodic measures related to pitch, speech rate, and rhythm from speakers with dysarthria and healthy controls in English and Korean datasets, despite the fact that these two languages differ in terms of prosodic characteristics. These prosody-based measures are then used as inputs to random forest, support vector machine and neural network classifiers to automatically assess different severity levels of dysarthria. Compared to baseline MFCC features, 18.13% and 11.22% relative accuracy improvement are achieved for English and Korean datasets, respectively, when including prosody-based features. Furthermore, most improvements are obtained with a better classification of mild dysarthric utterances: a recall improvement from 42.42% to 83.34% for English speakers with mild dysarthria and a recall improvement from 36.73% to 80.00% for Korean speakers with mild dysarthria.

**Keywords:** dysarthria; acoustics; prosody; detection; machine learning; support vector machine, random forest; neural network; feature selection

## 1. Introduction

Neurological disorders come with a range of cognitive and physical issues that can make life difficult for the affected individual. Speech is one aspect that can be severely damaged and lead to issues in communication. A common speech disorder known as dysarthria often occurs in individuals with a variety of neurological disorders such as Cerebral Palsy, Parkinson's Disease, Amyotrophic Lateral Sclerosis (ALS), Multiple Sclerosis, among others. Research indicates that dysarthria occurs up to 90% of the time in patients with Parkinson's Disease [1], and 50% of the time for individuals with Multiple Sclerosis [2]. Dysarthria is also one of the first symptoms of ALS in 25% of patients [3].

Dysarthria can also co-occur with aphasia in patients who suffered from acute stroke. The study in Reference [4] reports both dysarthria and aphasia in 10% of post-stroke patients, while in Reference [5], the co-occurrence reaches 29.6%. In Korean speakers, both dysarthria and aphasia are present in 35.4% of 150 post-acute stroke patients [6]. Dysarthria also occurs with non-fluent or agrammatic aphasia [7,8]. Researchers report dysprosody and a slower speech rate pose a prominent impairment in some patients with agrammatic aphasia [9,10]. Typically, dysarthria is diagnosed by a trained speech pathologist who administers several speaking tasks to the patient in order to perceptually evaluate any speech irregularities [11,12]. For example, one can determine impaired prosody by

having patients read sentences and observe any irregular variations in pitch, duration, or stress. Several standardized assessments based on perceptual evaluation have been proposed, with the Mayo Clinic Rating System [13] and Frenchay Dysarthria Assessment being the most detailed and commonly utilized tests for English speakers [14]. Despite the wide use of perceptual evaluation, the subjective nature of the task and excessively long duration of administering these tests are common criticisms. Furthermore, low identification accuracy is reported in Reference [15], while low intra- and inter-rater reliability are reported in Reference [16] and Reference [17] for the Mayo Clinic Rating System.

The prevalence of machine learning, especially deep learning, for speech and audio classification has introduced a variety of methods for automatically detecting and even assessing the severity of dysarthric speech. Detecting dysarthria involves extracting hand-crafted acoustic features and using those features as inputs to a machine learning-based classifier [18–20]. Deep learning approaches are also possible where the raw speech signal or a set of elementary features are fed into complex neural network architectures that automatically determine the important acoustic information and distinguish between healthy and dysarthric speech [21,22]. Deep learning approaches require less data preparation and feature engineering but may suffer from a lack of interpretability as further post-processing is often required to interpret how the speaker's speech is impaired. Another issue with the deep learning approach for impaired speech detection is the lack of data. The success of deep learning has largely been the result of big data and the ability to train on large datasets. Unfortunately, the collection of impaired speech data is difficult and available datasets are very limited.

Various types of acoustic features have been proposed for detecting dysarthric speech. Spectral features such as Mel Frequency Cepstral Coefficients (MFCCs) are used in References [22,23], and filter banks are utilized in long short-term memory classifiers [21] and convolutional neural networks [24]. Spectral measures of fricatives are shown to significantly differ between healthy and dysarthric speakers in Reference [25] and are used as input to a machine learning classifier in Reference [26]. Results from Reference [27] indicate that glottal features can improve detection performance against a baseline OpenSmile acoustic feature set.

The current study proposes measures related to prosody, as prosodic impairments are one of the earliest cues of dysarthria [28]. The study in Reference [28] examines the speech of 23 individuals with untreated early-stage Parkinson's, where 18 of the 23 are found to have dysarthria. Among the 18 individuals with dysarthria, phonatory deficits are present in 6 cases (26.09%), articulatory issues in 9 cases (39.13%), and 14 cases with prosody-based impairments (60.87%). Similarly, in Reference [29], features related to phonation (glottis features), articulation (MFCCs), and prosody (F0, energy, duration, pauses, jitter, and shimmer) are extracted from both healthy and dysarthric speakers with early-stage Parkinson's. These acoustic features are used as input to a support vector machine (SVM) classifier to predict whether a given utterance is from a healthy speaker or a speaker with dysarthria. Results show that training on prosodic features reaches an accuracy of 90.5%, while classifiers trained on glottal and articulatory features achieve an accuracy of 78.6% and 88.1%, respectively. Results from References [28,29] suggest that prosodic impairments are prevalent in dysarthric speech and prosody-based measures can be useful for the assessment of dysarthric speech.

The most salient prosodic impairments in dysarthric speech are related to pitch and speech rate. In Reference [30], the length of tone units, fundamental frequency (F0), and standard deviation of fundamental frequency from spontaneous speech are collected from healthy and dysarthric speakers. Speech from speakers with severe dysarthria displays shorter tone units and higher mean F0 compared to mild dysarthria and healthy controls. Patients with mild dysarthria have lower standard deviations of F0 (more monotonous speech) than healthy controls and severe dysarthric speakers. The findings in Reference [30] are further supported by later studies in speakers with multiple sclerosis, cerebral disease, and motor neuron disease [31,32]. Research with Korean speakers reports similar prosodic impairments in dysarthric speakers [33–35]. In Reference [33], an acoustic analysis of healthy speakers and speakers with dysarthria reveals longer syllable duration, varied pitch range, and more frequent pauses among speakers with dysarthria. Similarly, results from Reference [34] suggest that dysarthric

speech has lower pitch values for interrogative sentences. Lastly, the prosodic characteristics of speakers with dysarthria from a range of neurological disorders are analyzed in Reference [35]. Results indicate that dysarthric speech has a reduced speaking rate, reduced articulation rate, and reduced F0 slope compared to healthy speech.

Previous studies, such as in References [23,29], suggest that the use of prosody with machine learning methods can accurately classify healthy speech from dysarthric speech. In Reference [23], voice quality features such as HNR, shimmer, and jitter measures are extracted along with a prosody feature set, which includes F0 measures, utterance, and phone duration. A linear discriminant analysis (LDA)-based classifier is used to achieve an accuracy of 71.9% and 82.1% for voice quality and prosody feature sets, respectively. These previously mentioned studies take a binary approach to dysarthric speech detection and do not consider severity. However, in Reference [36], 11 prosodic features are used to automatically assess the severity level of dysarthric speakers from the publicly available Neymours database [37]. An LDA based feature selection method is used to determine the most discriminative prosodic features as follows (from the most to the least discriminative): articulation rate, number of pitch periods, mean pitch, voice breaks, %V, HNR, jitter, shimmer, standard deviation of pitch, standard deviation of pitch period, and NHR. These features can assess four levels of dysarthric speech with an accuracy of 88.89% when using a Gaussian mixture model classifier, and an accuracy of 93% with an SVM classifier. An issue with this study is the use of the Neymours database, which contains a limited number of speakers and stimuli. Only speech from male speakers is collected, and only 1 healthy control is available. Furthermore, the collected speech consists of the same carrier sentence, "The X is Y-ing the Z", which can lead to unnatural prosody. A close study that uses prosody-based features for severity assessment is in Reference [38], which focuses on rhythm metrics without including a baseline feature set of non-prosodic features. Results indicate a 15% and 3.2% relative accuracy increase in Korean and English datasets, respectively, compared to only using pitch, speech rate, and voice quality features.

Our study differs from the previously mentioned studies and contributes to the literature on dysarthric speech detection in several important aspects. First, our goal is to assess the severity of dysarthria with prosody-based measures; therefore, instead of a binary detection task where all dysarthric speakers are grouped into a single class, the speakers in our study are distinguished by four severity levels (healthy, mild, moderate and severe). In practical use, dysarthria assessment is most likely to be conducted on patients with early dysarthria, and therefore, it is important to include the speech of speakers with mild dysarthria. Second, we model prosody in a multidimensional manner by including measurements for pitch, speech rate, and rhythm. We also apply various feature selection algorithms to select measures that are more important for distinguishing different severity levels. Third, we conduct experiments with both English and Korean data, which allows for better generalization of using prosody with different languages. Furthermore, both databases contain a diverse set of speakers with multiple utterances from males, females, healthy, and dysarthric speakers.

It is important to examine languages with different prosodic systems as the efficacy of using prosody for dysarthria assessment may vary depending on the language. English is a lexical stress language with a variety of cues to signal stress [39]. Stressed syllables in English are perceptually louder than unstressed ones and tend to be realized higher in pitch and longer in duration. Vowels in unstressed syllables are reduced, resulting in changes of F1 and F2 formant frequencies. However, standard Korean (or Seoul Korean) does not have either lexical stress or lexical pitch accent. Only phrase-level tones are used in Korean [40]. The pitch of a Korean syllable is affected by the intonation pattern at the sentence level, given that the pitch accent is used to signify prosodic phrases in Korean. There are also rhythmic differences in Korean and English. English is typically described as a stress-timed language where the duration of stressed syllables is relatively equal [41]. Korean is commonly known as a syllable-timed language where all syllables tend to be equal in duration [42,43]. However, the status of rhythm in Korean is more contentious, and researchers have also proposed that Korean has stress-timed [44,45] and mora-timed [46] patterns.

The rest of the paper is organized as follows: Section 2 reviews the prosody-based measures, along with articulatory and phonatory features. Section 3 describes the experimental methods including the training process, classifiers, and feature selection methods. The Korean and English corpus is described in Section 4, along with an analysis of the prosodic measures. Results for all experiments are summarized in Section 5. Lastly, the study concludes in Section 6 with a discussion of the results and future directions.

## 2. Acoustic Measures

### 2.1. Prosody-Based Measures

The prosody-based features we use in our experiments are based on the findings from previous acoustic studies on dysarthric speech. In general, these values are irregular in speakers with dysarthria in comparison to the average healthy speakers. Prosodic features are grouped into 3 categories: pitch, speech rate, and rhythm. Compared to the study in Reference [36], which only uses 11 measures from the 3 prosodic categories, we include a wider range of 23 prosody-based measures as shown in Table 1. Pitch and speech rate are extracted using the Python library parselmouth, which provides a Pythonic interface to the internal Praat code [47]. Rhythm metrics are collected using the software Correlatore 2.3.4 [48].

**Table 1.** The full set of prosody-based measures.

| Pitch | Speech Rate | Rhythm |
|---|---|---|
| | | %V |
| F0 mean | # of syllables | Delta-V |
| F0 median | # of pauses | Delta-C |
| F0 min | Utterance duration | Varco-V |
| F0 max | Speaking duration | Varco-C |
| F0 std | Speaking rate | VrPVI |
| F0 quantile 25 | Articulation rate | CrPVI |
| F0 quantile 75 | Balance | VnPVI |
| | | CnPVI |

The acoustic representation of pitch is known as fundamental frequency (F0), which is the lowest frequency of a periodic waveform. F0 is measured for all voiced segments of an utterance. The pitch feature set includes mean, median, minimum, and maximum F0 along with the standard deviation, 25%, and 75% quantiles.

The speech rate refers to the speed-related measurements of speech such as speaking rate (syllables per second) and articulation rate (syllables/per second without pause). The current study includes seven speech rate measures per utterance: number of syllables, number of pauses, full utterance duration, speaking duration (excluding pauses), speaking rate, articulation rate, and balance. Balance refers to the ratio of speaking duration to utterance duration.

In linguistics, rhythm corresponds to the duration-based division of speech units into relatively equal pieces. Duration-based measures of vocalic and intervocalic segments are proposed as correlates of rhythm in the speech signal [49–51]. The first group of 3 rhythm metrics is proposed in Reference [49]: the proportion of vocalic intervals (%V) and the standard deviations of consonantal ($\Delta$C) and vocalic ($\Delta$V) intervals.

In Reference [50], the influence of speech rate on $\Delta$C and $\Delta$V is addressed. Delta values can be normalized by dividing the delta into the mean duration of vocalic or consonantal intervals, then multiplying that by 100. The normalized delta values are known as the 'Varcos' and can be

measured for both vowel and consonant intervals. Varco-C and Varco-V can be calculated as shown in Equations (1) and (2), respectively:

$$\text{Varco} - \text{C} \;=\; \frac{\Delta C * 100}{mean(C)} \tag{1}$$

$$\text{Varco} - \text{V} \;=\; \frac{\Delta V * 100}{mean(V)} \tag{2}$$

The next group of rhythm metrics is calculated using the pairwise variability index (PVI), where the temporal succession of the vocalic or consonantal intervals are taken into consideration [51]. The influence of speech rate variation can be controlled by calculating the normalized PVI, which calculates the mean absolute normalized difference between durations of neighboring interval pairs. Both rPVI and nPVI are defined in Equations (3) and (4), where $d_k$ is the length of the $k$th vocalic or consonantal segment and $m$ is the number of segments. The raw PVI can be calculated for vowel intervals (VrPVI) or consonant intervals (CrPVI). Similarly, the normalized PVI can also be calculated for vowel intervals (VnPVI) or consonant (CnPVI) intervals.

$$\text{rPVI} \;=\; \sum_{k-1}^{m-1} \left| d_k - d_{k+1} \right| / (m-1) \tag{3}$$

$$\text{nPVI} \;=\; 100 * \sum_{k-1}^{m-1} \left| \frac{d_k - d_{k+1}}{\frac{d_k + d_{k+1}}{2}} \right| / (m-1) \tag{4}$$

### 2.2. Articulation and Phonation Based Measures

MFCCs are commonly used in automatic speech recognition (ASR) as they represent the relevant frequencies shaped by the vocal tract while removing irrelevant F0 information. A baseline set of acoustic features representing articulation (MFCCs) and phonation (voice quality) are extracted for comparison of prosodic features with non-prosodic features. We extract 12-dim MFCCs and log energy along with delta and delta-delta features from all utterances. This process leads to a 39-dim feature set representing articulation.

Voice quality refers to the properties of speech related to the vocal folds within the larynx. Our study includes the following voice quality measures, which are used in acoustic studies of dysarthric speech [52,53]: relative jitter, relative shimmer, harmonics to noise ratio (HNR), # of voice breaks, and degree of voice breaks. Some studies, such as References [29,36], include jitter and shimmer in prosody-based feature sets; however, these features are related to phonation, and thus, are better categorized as voice quality features. Therefore, we consider jitter and shimmer as voice quality features similar to the studies in Reference [23,28]. All voice quality measures are extracted with the Praat software using the voice analysis function [54].

Jitter represents the variations of F0 within a time period. We calculate relative local jitter by the average absolute difference between consecutive periods, divided by the average period. The calculation for jitter can be examined in Equations (5)–(7), where $T_i$ is the duration of the $i$th interval and $N$ is the number of intervals.

$$\text{Absolute jitter (sec)} \;=\; \sum_{i=1}^{N} \left| T_i - T_{i+1} \right| (N-1) \tag{5}$$

$$\text{Mean Period (sec)} \;=\; \sum_{i=1}^{N} T_i / N \tag{6}$$

$$\text{Relative Jitter = Absolute Jitter/Mean Period} \tag{7}$$

Shimmer, as shown in Equation (8), where $A_i$ is the amplitude of the *i*th interval, is very similar to jitter, except that F0 perturbations fall in the amplitude domain; therefore, the average absolute difference between the amplitudes of consecutive periods are divided by the average amplitude.

$$\text{Relative Shimmer} \;=\; \frac{\frac{1}{N-1}\sum_{i=1}^{N-1}|A_i - A_{i=1}|}{\frac{1}{N}\sum_{i=1}^{N} A_i} \tag{8}$$

HNR refers to the periodicity of a speech signal over noise. Harmonicity is measured in decibels (dB) by the ratio of the energy of the periodic part ($E_p$) related to the noise energy ($E_n$) as shown in Equation (9).

$$\text{HNR (dB)} \;=\; 10\log\left(\frac{E_p}{E_n}\right) \tag{9}$$

Lastly, we include two measures related to breaks in voicing. The first is the number of voice breaks which Praat calculates by taking the number of distances between consecutive glottal pulses that are longer than 1.25 divided by the pitch floor which is set at 50 Hz. Any interval longer than 25 ms between pulses is considered a voice break. Second, we measure the degree of voice breaks, which is the total duration of the breaks in the signal, divided by the total duration, excluding silence at the beginning and the end of the sentence.

## 3. Methods

### 3.1. Dysarthria Severity Assessment

A visualization of the assessment process is shown in Figure 1. The process of the experiments is as follows: extract acoustic features from all utterances, select optimal features via feature selection algorithms, split data into train and test sets, and lastly, feed features to machine learning classifiers. The classifiers are random forest (RF), support vector machine (SVM), and neural networks (NN). The RF algorithm is a robust method for classification with small datasets. An RF classifier is less influenced by outliers and handles noisy data well. SVMs are commonly used classifiers in machine learning, and in particular for impaired speech detection [23,26,27,29]. The success of SVMs is not limited to early studies but continues to show good performance even in recent studies as they consistently perform well, even with small datasets [18–20]. Lastly, we include NNs, as they have shown state-of-the-art results for many speech and audio classification tasks.

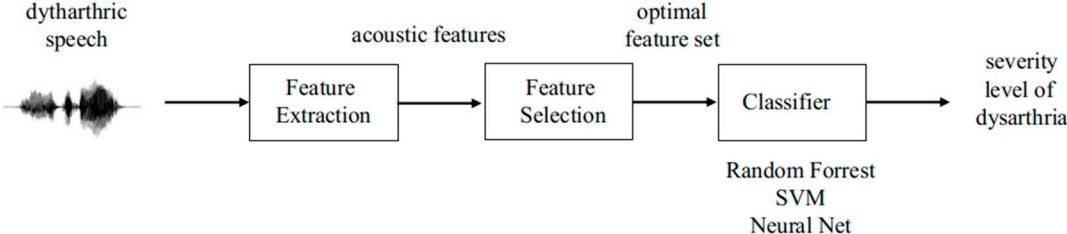

**Figure 1.** Block diagram of dysarthria severity assessment using acoustic features.

We use an SVM with a radial basis kernel function and optimized the C and gamma parameters via a grid search using values between $10^{-4}$ and $10^4$. For the RF classifier, the number of trees and the depth of trees are optimized. A forest with 100 trees led to the highest accuracy, while the optimal depth of trees is 30. Lastly, we test the performance of different NN classifiers with different parameters as shown in Table 2. After hyperparameter optimization, the NN classifier which led to the highest accuracy was one that contained two hidden layers and a ReLU activation function. The neural network is trained with an Adam optimizer which yields better results than standard stochastic gradient descent or limited-memory BFGS solvers. An adaptive learning rate is initialized with a value of 0.001.

**Table 2.** The different parameters examined with NN classifiers.

| Parameters | # of Hidden Layers | # of Neurons | Activation Function | Optimizer | Learning Rate |
|:---:|:---:|:---:|:---:|:---:|:---:|
| Values | 1, 2, 3, 4 | 25–100 | Logistic, tanh, ReLU | Adam, SGD, LBFGS | 0.0001–0.1 |

Along with training our classifiers using the features selected from the feature selection algorithms, we also train on individual acoustic groups. Furthermore, results are compared with baseline MFCC and voice quality features. Choosing the right set of features is an important aspect when training machine learning models. In order to select the optimal set of prosodic features, we conduct several feature selection methods and compare the performance of each method. Section 3.2 describes the different methods used for the current study.

*3.2. Feature Selection*

Three major feature selection methods for machine learning are examined: filter, wrapper, and embedded methods [55–57]. The filter method is the most computationally efficient but is unable to handle redundant features. Furthermore, important features that are less informative on their own but are informative when combined with others may be discarded. The wrapper method tends to produce better classification accuracy, but it is the most computationally complex method and does not generalize well to other datasets. The embedded approach tries to alleviate the computation time required by the wrapper method by incorporating the feature selection process with the overall training process. While the embedded approach is computationally less expensive than the wrapper method, it still has the issue of generalizability.

The filter method works by selecting the best features based on univariate statistical tests. The selection of features is independent of any machine learning algorithm. Features are ranked based on statistical scores which determine the correlation of features with the outcome variable. In our case, we use ANOVA F-values since our groups are categorical. F-values are the ratios of two Chi-distributions divided by its degrees of freedom, as shown in Equation (10).

$$\text{F} = \left( \chi_1^2 / n1 - 1 \right) / \left( \chi_2^2 / n2 - 1 \right) \tag{10}$$

Next, we test two embedded feature selection methods, an L1-based (lasso) feature selection and a tree-based feature importance method. Tree-based estimators are used to compute impurity-based feature importance values, which in turn are used to discard irrelevant features. The lasso method is a regularization approach where a penalty is applied over the coefficients of a linear model. This is accomplished by modifying the mean squared error cost function to contain the L1-regularization ($\lambda$), as shown in Equation (11). Features with non-zero coefficients are selected for model training.

$$\text{J}(\theta) = \frac{1}{m} \sum_{i=1}^{m} Cost\left( h_\theta\left( x^i \right), y^i \right) + \frac{\lambda}{m} \sum_{j=1}^{n} \left| \theta_j \right| \tag{11}$$

The wrapper method of feature selection used is Recursive Feature Elimination (RFE). RFE performs a greedy search to find the best performing feature subset. It iteratively creates models and determines the best or the worst performing feature at each iteration, then constructs the subsequent models with the leftover features until all the features are explored. The current study uses a linear SVM as the model evaluator. RFE then ranks the features based on the order of their elimination.

## 4. Materials

*4.1. English Corpus (TORGO)*

Two commonly utilized datasets for English dysarthric speech research are the UA-Speech Database [58] and the TORGO database [59]. The former is larger and only contains isolated words, while the latter is composed of both isolated words and continuous speech. Isolated words may be

sufficient to detect dysarthria from speakers with severe or moderate dysarthria but may not be enough input to detect mild dysarthria. Furthermore, prosody is a dynamic aspect of speech which is better represented by continuous speech. Therefore, we only use continuous speech data in our experiments to check the effectiveness of prosody-based measures.

The TORGO dataset contains 8 speakers with dysarthria, 5 males and 3 females from patients with Cerebral Palsy, and 1 speaker with ALS. Speakers are recruited by a speech pathologist in Toronto Ontario, Canada. While the exact dialect is not mentioned in Reference [59], speakers are likely to have a standard Canadian English dialect. Dialect consistency may be an important factor as different dialects can display different prosodic characteristics. Since we are using prosody-based measures as features, we want the machine learning model to detect prosodic variation between healthy speech and dysarthric speech not between dialects.

Speakers with dysarthria are assessed by a trained speech pathologist using the Frenchay Dysarthria Assessment [7]. Two speakers are categorized as having severe dysarthria, one speaker with moderate/severe, one moderate, and four mild. Recording from 7 healthy controls, 4 males and 3 females, are also collected. A mixture of short words, non-words, restricted sentences (read speech), and unrestricted sentences (spontaneous speech) are recorded from all speakers, but only restricted sentences are analyzed for our study. Some examples of recorded speech stimuli can be examined in Table A1 from the Appendix A.

### 4.2. Korean Corpus (QoLT)

For Korean, we use the Quality of Life Technology (QoLT) database [60]. The QoLT database contains recordings from 100 dysarthric speakers and 30 healthy controls. Researchers collected demographic information regarding the place of growth before the age of 12. Most speakers are from Seoul or the Gyeonggi province; therefore, they are likely to speak with a standard Seoul dialect. However, one speaker in our study grew up in the Gyeongsang province, which is known to be the location of a pitch-accent dialect. A group of speech therapists assessed the severity of speakers via Percentage of Consonant Correctness (PCC) using the Assessment of Phonology and Articulation for Children (APAC) words. A subset of assessments was re-evaluated and resulted in an intra-rater reliability of 0.957 and inter-rater reliability of 0.901 using Pearson's product moment correlation. Several types of speech stimuli were recorded from all speakers, including machine control commands, phonetically balanced words and sentences for investigating a variety of Korean consonants and vowels in different phonetic environments. As with the TORGO dataset, only data from continuous speech samples were used in our experiments (see Table A2 in the Appendix A for the full set of sentences).

### 4.3. Data Analysis

A data analysis of prosody-based features and voice quality is conducted to examine any significant difference among severity groups. Furthermore, the data analysis allows us to examine if the TORGO and QoLT datasets follow the findings of previous acoustic studies on dysarthric speech. Along with presenting mean values for all prosody-based measures, we implement a two-way mixed ANOVA with the severity group as the between-subject variable and individual prosodic measures as the within-subject variable. Post-hoc tests are conducted to examine any significant findings with simple main effects, and if a main effect exists, we further implement multiple pairwise comparisons to determine which severity groups are different.

#### 4.3.1. Pitch

Mean values of all F0 measurements for English and Korean speakers are shown in Figures 2 and 3, respectively. In both TORGO and QoLT datasets, speakers with dysarthria generally have higher F0 values. Even English speakers with mild dysarthria tend to have a higher F0, except in the case of maximum F0. Furthermore, English speakers with mild dysarthria have a lower standard deviation (25.35 Hz) compared to healthy speakers (35.5 Hz), supporting the findings from [30] that speakers with

mild dysarthria tend to be more mono-pitch. However, the opposite is apparent in Korean speakers where healthy speakers have a slightly lower standard deviation (30.2 Hz) compared to the mild group (35.2 Hz).

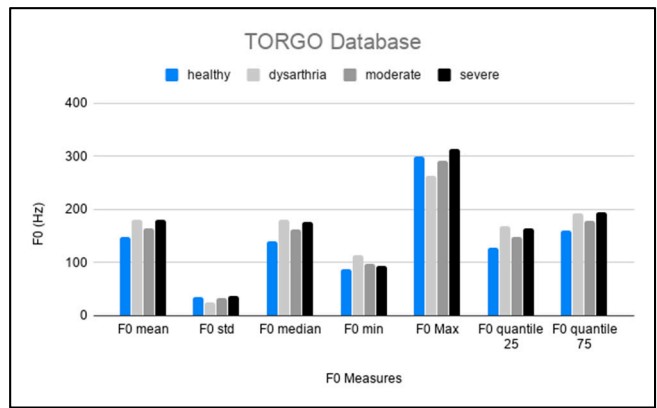

**Figure 2.** The mean values of all F0 measurements from the TORGO dataset.

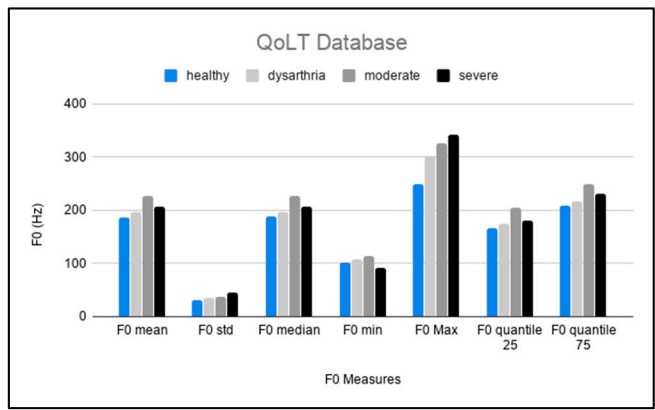

**Figure 3.** The mean values of all F0 measurements from the QoLT dataset.

A two-way mixed ANOVA revealed no significant two-way interaction between the severity of the group and pitch measure for English, where $F_{(5.33, 19.56)} = 0.59$, $p > 0.05$, but a significant interaction existed with Korean speakers, where $F_{(6.31, 69.38)} = 2.90$, $p < 0.05$. Note when calculating F-values, the values in brackets refer to the degree of freedom for between-subject and within-subject variables. Furthermore, a post-hoc test with a Bonferroni adjusted $p$-value revealed that the simple main effect is significant for max F0, where $F_{(3, 33)} = 2.02$, $p < 0.05$.

### 4.3.2. Speech Rate

Dysarthric speakers tend to have prolonged pronunciation leading to a slower speaking rate, slower articulation rate, more pauses, and more syllables. Impairments in speech rate are apparent when examining the mean values from Table 3. The difference in speech rate measures between healthy and dysarthric speakers is larger in Korean than in English. For example, the number of pauses for healthy and severe dysarthric speakers is 0.17 and 1.82 for English speakers, respectively, whereas, 0.09 and 4.74 for Korean speakers, respectively.

According to the results from the two-way mixed ANOVA, there is no significant interaction between severity group and speech rate measure for English. However, a post-hoc pairwise comparisons test reveals a significant difference between healthy and severe speakers for articulation rate and speaking duration ($p < 0.05$). Unlike English, a significant two-way interaction is present for Korean utterances, where $F_{(8.42, 92.6)} = 12.83$, $p < 0.05$. Furthermore, the simple main effect of severity on all speech rate measures except for balance is significant ($p < 0.05$). Pairwise comparisons tests reveal

significant differences between healthy and severe utterance for all speech rate measures except balance ($p < 0.05$), between healthy and moderate utterances for all measures except balance and number of syllables ($p < 0.05$), between healthy and mild utterances for speaking rate ($p < 0.05$). Significant differences are also apparent with mild and severe utterances for all measures except articulation rate and balance ($p < 0.05$), and between moderate and severe utterances for the number of syllables ($p < 0.05$).

**Table 3.** Mean values for all speech rate measures from the QoLT and TORGO datasets.

| Corpus | Speaker Group | # of Syllables | # of Pauses | Speaking Rate | Artic. Rate | Speaking Duration (s) | Total Duration (s) |
|---|---|---|---|---|---|---|---|
| TORGO | Healthy | 9.12 | 0.17 | 2.00 | 4.14 | 2.22 | 4.50 |
| | Mild | 10.67 | 1.40 | 1.75 | 3.53 | 3.10 | 6.22 |
| | Moderate | 10.78 | 2.09 | 1.78 | 3.33 | 3.38 | 6.60 |
| | Severe | 12.21 | 1.82 | 1.69 | 3.17 | 3.77 | 7.13 |
| QoLT | Healthy | 11.75 | 0.09 | 3.00 | 4.58 | 2.52 | 3.83 |
| | Mild | 13.14 | 1.45 | 2.29 | 3.82 | 3.58 | 6.00 |
| | Moderate | 13.91 | 3.29 | 1.69 | 3.42 | 4.19 | 8.96 |
| | Severe | 17.53 | 4.74 | 1.64 | 3.47 | 5.29 | 11.49 |

### 4.3.3. Rhythm

In general, English speakers with dysarthria have higher values than healthy speakers for most cases except %V and Varco-V as shown in Table 4. With the exception of %V in mild speakers, all Korean speakers with dysarthria have higher values for both vowel and consonant measures. Furthermore, the differences between healthy and dysarthric speech are larger in Korean speakers than English speakers. For example, the standard deviation of vowel intervals (Delta-V) is 60.70 and 66.23 in healthy English and mild dysarthric speakers, respectively, while the same measure is 65.69 and 98.75 for Korean healthy and mild dysarthric speakers, respectively.

**Table 4.** Mean values for all rhythm metrics from TORGO and QoLT datasets.

| Speaker | %V | Delta-V | Delta-C | Varco-V | Varco-C | VrPVI | CrPVI | VnPVI | CnPVI |
|---|---|---|---|---|---|---|---|---|---|
| TORGO-healthy | 41.72 | 60.70 | 73.28 | 53.18 | 50.89 | 66.20 | 81.85 | 55.85 | 56.89 |
| TORGO-mild | 42.50 | 66.23 | 72.56 | 54.18 | 50.00 | 71.14 | 75.01 | 57.67 | 52.89 |
| TORGO-mod | 41.78 | 94.23 | 104.32 | 47.03 | 50.92 | 105.87 | 113.87 | 50.48 | 55.48 |
| TORGO-severe | 46.23 | 121.06 | 147.46 | 50.24 | 64.21 | 133.62 | 162.25 | 53.40 | 67.41 |
| QoLT-healthy | 54.37 | 65.69 | 51.79 | 57.59 | 55.23 | 67.52 | 65.78 | 61.51 | 70.36 |
| QoLT-mild | 54.06 | 98.74 | 77.74 | 59.94 | 57.26 | 101.99 | 93.62 | 61.89 | 69.06 |
| QoLT-mod | 56.31 | 120.13 | 93.47 | 56.58 | 59.62 | 126.75 | 105.42 | 59.95 | 67.26 |
| QoLT-severe | 64.71 | 217.85 | 125.35 | 58.88 | 64.22 | 237.38 | 139.24 | 60.90 | 71.84 |

Rhythm metrics are the only group of prosody-based measures to have a significant two-way interaction for both English, where $F_{(4.77, 17.49)} = 8.21$, $p < 0.0001$, and Korean datasets, where $F_{(4.13, 45.43)} = 8.05$, $p < 0.0001$. With the English TORGO dataset, a simple main effect of the severity group is significant for CrPVI, Delta-C, Delta-V, and Varco-C ($p < 0.05$). The mean rhythm score is significantly different in healthy vs. severe and mild vs. severe pairs for CnPVI, CrPVI, Delta-C, and Varco-C ($p < 0.05$). There is also a significant difference in Delta-V between healthy and severe utterances ($p < 0.05$). Lastly, there is a significant difference in Varco-C between moderate and severe utterances ($p < 0.05$). For the Korean QoLT dataset, simple main effects are significant for VrPVI, CrPVI, Delta-C, Delta-V, %V and Varco-C. Pairwise comparisons tests reveal significant differences between healthy and severe utterances for CrPVI, Delta-C, Delta-V, and %V ($p < 0.05$), and between healthy and moderate utterances for CrPVI and Delta-C ($p < 0.05$). Significant differences are also present between mild and severe utterances for all measures, except CnPVI and Varco-V ($p < 0.05$), and between moderate and severe utterances for Delta-C, Delta-V, and %V ($p < 0.05$).

### 4.3.4. Voice Quality

The mean values for all voice quality measures in Korean and English are shown in Table 5. In general, English speakers with dysarthria are observed to have higher values for all measures except shimmer, while Korean dysarthric speakers have higher values for all measures except jitter and shimmer. Results from the two-way mixed ANOVA suggest no significant two-way interaction for the English TORGO dataset, but a significant interaction for the Korean QoLT dataset, where F $(5.06, 55.67) = 20.01$, $p < 0.05$. Furthermore, the main effect of the severity group and voice quality measure is significant for all measures ($p < 0.05$). Pairwise comparisons tests further reveal that significant differences depend on the specific measure and severity group. A significant difference is seen between healthy vs. severe and healthy vs. moderate utterances for the degree of voice break and the number of voice breaks ($p < 0.05$), between healthy and mild utterances for jitter, shimmer, and mean HNR ($p < 0.05$). Significant differences are also apparent between mild and moderate utterances for all measures except the number of voice breaks ($p < 0.05$), between mild and severe for all measures ($p < 0.05$), and between moderate and severe for mean HNR ($p < 0.05$).

**Table 5.** Mean values for voice quality measures from TORGO and QoLT datasets.

| Corpus | Speaker Group | Jitter | Shimmer | HNR | # of VB | % of VB |
|---|---|---|---|---|---|---|
| TORGO | Healthy | 1.85 | 11.46 | 9.59 | 6.00 | 17.13 |
| | Mild | 2.02 | 9.76 | 10.23 | 6.70 | 16.71 |
| | Moderate | 1.80 | 7.98 | 13.75 | 7.80 | 27.10 |
| | Severe | 2.24 | 8.46 | 12.67 | 9.29 | 20.86 |
| QoLT | Healthy | 1.68 | 7.54 | 15.12 | 5.71 | 13.15 |
| | Mild | 1.53 | 7.08 | 15.78 | 7.89 | 20.90 |
| | Moderate | 1.62 | 6.94 | 16.14 | 9.30 | 33.39 |
| | Severe | 1.69 | 7.37 | 15.50 | 11.84 | 34.58 |

### 4.3.5. Data Analysis Summary

There are significant differences between healthy and severe utterances for articulation rate, speaking duration, CnPVI, CrPVI, Delta-C, Delta-V, and Varco-C in the TORGO dataset. The rhythm metrics CnPVI, CrPVI, Delta-C, and Varco-C significantly differ between mild and severe utterances. Lastly, moderate and severe utterances have significantly different Varco-C measures. The TORGO dataset lacks statistically significant differences for pitch and voice quality measures. Based on the results from mixed ANOVA tests, there are few statistically significant differences in the TORGO dataset compared to the QoLT dataset, but this may be caused by the low number of speakers in TORGO (15 persons) compared to the number of speakers from QoLT (38 persons).

More significant differences exist for the QoLT dataset. Measures with significant differences between healthy and severe utterances are as follows: max F0, all speech rate measures (except balance), CrPVI, Delta-C, Delta-V, %V, degree of voice break, and the number of voice breaks. Significant differences exist between healthy and moderate utterances for the following measures: number of pauses, utterance duration, speaking duration, rate of speech, articulation rate, CrPVI, Delta-C, degree of voice break, and number of voice breaks. Between healthy and mild utterances, there are significant differences in speaking rate and all voice quality measures. Measures with significant differences between mild and severe utterances are as follows: all speech rate measures except articulation rate and balance, all rhythm measures except CnPVI and Varco-V, and all voice quality measures. Only voice quality measures have significant differences between mild and moderate utterances. Lastly, significant differences exist between moderate and severe utterances for the following measures: number of syllables, Delta-C, Delta-V, %V, and mean HN.

*4.4. Data Organization*

For the TORGO dataset, restricted sentences (read speech) from all healthy and dysarthric speakers are included in our experiments. We train on 340 utterances split between speaker groups. This training set is used during hyperparameter tuning where a 10-fold cross-validation is implemented. After selecting the optimal parameters for the classifiers, we test model performance on 238 separate utterances. Given the small number of speakers in the TORGO dataset, the training and test sets are split by unique sentences such that no sentence in the training set is present in the test set. This leads to speaker-dependent classifiers.

The QoLT dataset contains many more speakers but fewer sentences; therefore, the training and test sets are split between speakers such that no speaker in the training set is present in the test set. Therefore, unlike the TORGO dataset, we build speaker-independent classifiers. The training set contains 6 healthy speakers and 6 dysarthric speakers for each severity level except 'severe', which has 5 speakers. The test set contains 4 different speakers for the healthy group and 4 different speakers for each dysarthric severity group. In total, 230 utterances are used for training and 117 for testing.

## 5. Results

The performance of classifiers is evaluated using accuracy, precision, recall, and F1-score. Accuracy is calculated when examining the overall performance of classifiers, while precision, recall, and F1-score are used when evaluating the performance of individual severity groups. The accuracy refers to the sum of true positives for all severity groups, including the healthy group, divided by the total number of utterances. Precision refers to the number of true positives divided by the sum of true positive and false positives. Recall refers to the number of true positives divided by the sum of true positives and false negatives. All classifiers, feature selection algorithms, and evaluation metrics are implemented using the Sci-Kit Learn machine learning library for Python [61].

*5.1. Feature Selection*

The method of feature selection is important for finding the set of prosody-based features that produce the highest accuracy. According to the results from Table 6, RFE selected features are optimal for the TORGO dataset, with an accuracy of 66.39%. The tree-based selected features are optimal for the QoLT dataset and lead to an accuracy of 66.67%. Furthermore, the exact features selected for the TORGO and QoLT datasets are shown in Table 7. In total, 14 features are optimal for TORGO, while 21 are optimal for QoLT. Based on the results from Table 8, classifiers trained on only prosody-based selected features reach an accuracy of 66.39% and 66.67% for TORGO and QoLT datasets, respectively, which is a higher accuracy compared to classifiers trained on MFCC or voice quality features alone. Results from Tables 6 and 8 represent the accuracy of the best performing classifier, which is an NN for TORGO and an RF for QoLT.

**Table 6.** Accuracy of different feature selection algorithms using prosody-based features.

| Feature Selection Method | TORGO Accuracy % | QoLT Accuracy % |
|---|---|---|
| Filter Method | 64.29 | 65.81 |
| Tree-Based | 64.71 | 66.67 |
| Lasso-Based | 63.87 | 62.39 |
| RFE | 66.39 | 63.25 |
| All | 61.76 | 64.96 |

**Table 7.** Prosody-based features selected for TORGO and QoLT datasets.

| Prosodic Group | Optimal Features for TORGO | Optimal Features for QoLT |
|---|---|---|
| Pitch | F0 mean, F0 median, F0 quantile 25, F0 quantile 75 | F0 median, F0 quantile 75, F0 quantile 25, F0 max, F0 min, F0 mean, F0 std |
| Speech Rate | number of pauses, speaking duration, total duration, balance | total duration, number of pauses, speaking duration, balance, speaking rate |
| Rhythm | Delta-V, Delta-C, Varco-C, VrPVI, CrPVI, VnPVI | %V, VrPVI, VnPVI, Varco-C, CnPVI, Varco-V, Delta-C, Delta-V, CrPVI |

**Table 8.** Severity assessment accuracy scores with acoustic measures.

| Features | TORGO Accuracy % | QoLT Accuracy % |
|---|---|---|
| Prosody | 66.39 | 66.67 |
| MFCC | 64.02 | 60.00 |
| Voice Quality | 53.82 | 58.76 |

### 5.2. TORGO

Compared to combining MFCCs with voice quality or prosody with voice quality features, higher improvements with all three classifiers are obtained when combining MFCCs with prosody features, as shown in Table 9. In comparison to solely using MFCC features, there is an improvement in accuracy from 64.02% to 75.63% when combining MFCCs with prosody-based features using an NN classifier. Table 10 reveals that the majority of improvements with the NN model is achieved with the better classification of mild utterances. When including prosody-based measures, a recall improvement from 42.42% to 83.34% and a precision improvement from 68.29% to 80.90% are achieved for the classification of mild utterances.

**Table 9.** Accuracy results for various acoustic feature combinations.

| Classifier | Accuracy % MFCC + Prosody | Accuracy % MFCC + Voice Quality | Accuracy % Prosody + Voice Quality | Accuracy % All Acoustic Features |
|---|---|---|---|---|
| RF | 60.50 | 57.98 | 58.40 | 62.18 |
| SVM | 68.91 | 67.65 | 68.07 | 71.00 |
| NN | 75.63 | 69.33 | 65.97 | 73.89 |

**Table 10.** Evaluation metrics for all severity groups using an NN classifier.

| Features | Evaluation Metrics | Healthy | Mild | Moderate | Severe |
|---|---|---|---|---|---|
| Only MFCC | F1-score % | 84.30 | 52.34 | 50.32 | 73.68 |
|  | Precision % | 82.26 | 68.29 | 70.91 | 59.32 |
|  | Recall % | 86.44 | 42.42 | 39.00 | 97.22 |
| MFCC + Prosody | F1-score % | 86.23 | 82.11 | 57.14 | 68.02 |
|  | Precision % | 82.81 | 80.90 | 52.33 | 82.92 |
|  | Recall % | 89.84 | 83.34 | 63.00 | 57.61 |

### 5.3. QoLT

Results from the QoLT dataset indicate higher accuracy when training on all acoustic features as shown in Table 11. However, compared to solely using MFCCs, there is still an accuracy improvement from 60.00% to 66.73% when including prosody-based features with an RF classifier. As shown in Table 12, most improvements are with mild and healthy utterances. There is a recall increase from 36.73% to 80.00% and a precision increase from 52.52% to 66.67% for mild utterances. Furthermore, healthy utterances are more accurately classified when including prosody-based features and reach 100% for all evaluation metrics. This implies no healthy utterance is misclassified as dysarthric and no dysarthric utterance is misclassified as healthy.

**Table 11.** Accuracy results for various acoustic feature combinations.

| Classifier | Accuracy % MFCC + Prosody | Accuracy % MFCC + Voice Quality | Accuracy % Prosody + Voice Quality | Accuracy % All Acoustic Features |
|---|---|---|---|---|
| RF | 66.73 | 64.11 | 60.73 | 70.10 |
| SVM | 65.78 | 66.67 | 64.11 | 67.52 |
| NN | 56.40 | 54.72 | 57.78 | 61.51 |

**Table 12.** Evaluation metrics for all severity groups when using an RF classifier.

| Features | Evaluation Metrics | Healthy | Mild | Moderate | Severe |
|---|---|---|---|---|---|
| Only MFCC | F1-score% | 90.00 | 43.10 | 48.6 | 49.11 |
| | Precision % | 87.45 | 52.52 | 42.53 | 51.99 |
| | Recall % | 93.37 | 36.73 | 56.70 | 46.72 |
| MFCC + Prosody | F1-score % | 100.00 | 72.78 | 55.11 | 36.04 |
| | Precision % | 100.00 | 66.67 | 63.32 | 33.33 |
| | Recall % | 100.00 | 80.00 | 48.74 | 39.11 |

## 6. Discussion & Conclusions

In the current study, we examine the use of prosody-based features for severity assessment of dysarthric speech for both English and Korean datasets. According to the results from machine learning experiments, higher accuracy is achieved when using prosody-based measures in comparison to MFCC or voice quality features. In particular, prosody is effective for improving the classification of mild dysarthria. This is promising in showing that even mild impairments in speech can be automatically detected. Our findings support and further generalize the results from Reference [29]. The best performance with the TORGO dataset occurs when training on both MFCCs and prosody-based features. An accuracy of 75.63% is obtained using a neural network classifier, which is a relative accuracy increase of 18.13% in comparison to solely using baseline MFCC features (64.02% accuracy). For the QoLT dataset, a random forest classifier trained on all acoustic features leads to the highest accuracy of 70.10%, which is a relative accuracy increase of 16.83% compared to solely training on MFCCs (60.00% accuracy). Furthermore, the classification of mild dysarthric utterances is improved when including prosody-based features. For English, this leads to a recall improvement from 42.42% to 83.34% for mild utterances, and for Korean, a recall increase from 36.73% to 80.00%.

Overall accuracy (75.63% for English and 70.10% for Korean) for both datasets is lower than results from previous studies, but important differences may explain why this may be the case. In Reference [36], researchers also use prosody-based features for severity assessment and reach an accuracy of 93%, but as previously mentioned, the database used in the study only contains male speakers, only one healthy speaker, and only one type of non-sense carrier sentence. To increase generalizability, our study uses a different database that contains several utterances produced by multiple males, females, healthy, and dysarthric speakers. Aside from the study in Reference [38], no other study examines the use of prosody-based measures for severity assessment with the TORGO dataset. However, there are studies using other acoustic features. An accuracy of 98.7% was achieved in Reference [62], using timbre-based features, but this study does not include healthy speakers. Furthermore, isolated words are also considered in their dataset, whereas we only considered continuous speech. Similarly, the study in Reference [63] uses MFCCs along with delta features in a CNN classifier with the TORGO dataset and reaches a severity assessment accuracy of 98.3%. However, this study also excludes healthy speech data and only conducts testing on two speakers (one mild and one severe), whereas the current study examines all speakers for testing.

In a clinical setting, it would be helpful to not only assess dysarthria but also give diagnostics of specific speech impairments. While spectral information can help detect dysarthria, they can be difficult to interpret and, in some cases such as MFCCs, remove prosodic information. By including prosody-based features, clinicians can better interpret what aspects of speech are impaired and prepare

appropriate therapy. Future studies should compare results when using prosody-based features in other forms of dysarthria such as Parkinsonian dysarthria to provide stronger generalization. A cross-database analysis to confirm our findings with other datasets will also be required to further support the benefits of prosody for dysarthria severity assessment. Lastly, a follow-up study on how prosody-based features can be used for diagnosis would be important to help speech therapists determine what specific aspects of prosody are impaired in an individual with dysarthria.

**Author Contributions:** Conceptualization, A.H. and M.C.; methodology, A.H.; software, A.H.; validation, S.K., M.C.; formal analysis, A.H.; investigation, A.H.; resources, A.H., M.C.; data curation, A.H., S.K., M.C.; writing—original draft preparation, A.H.; writing—review and editing, A.H., S.K., M.C.; visualization, A.H.; supervision, S.K., M.C.; project administration, M.C.; funding acquisition, M.C. All authors have read and agreed to the published version of the manuscript.

**Funding:** This research is supported by the Ministry of Culture, Sports and Tourism (MCST) and Korea Creative Content Agency (KOCCA) in the Culture Technology (CT) Research & Development Program 2020.

**Conflicts of Interest:** The authors declare no conflict of interest.

## Appendix A

**Table A1.** Speech stimuli examples from the TORGO database.

| Type of Sentence | Examples |
|---|---|
| Restricted Sentences | • Preselected phoneme-rich sentences such as:<br>　○　"The quick brown fox jumps over the lazy dog"<br>• The Grandfather passage<br>• 162 sentences from the sentence intelligibility section of the Yorkston Beukelman Assessment of Intelligibility of Dysarthric Speech<br>• The 460 TIMIT-derived sentences used as prompts in the MOCHA database |

**Table A2.** Continuous speech stimuli recorded for the QoLT database.

| Korean Hangul | Yale Romanization | English Translation |
|---|---|---|
| 추석에는 온 가족이 함께 송편을 만든다. | chwusekeynun on kacoki hamkkey songphyenul mantunta | In Chuseok, the whole family makes songpyeon together. |
| 갑자기 미국에 있는 오빠 얼굴이 보고 싶다. | kapcaki mikwukey issnun oppa elkwuli poko siphta. | Suddenly, I want to see my brother's face who is in America. |
| 어제 하늘이 컴컴해지더니 비가 쏟아졌다. | ecey hanuli khemkhemhayciteni pika ssotacyessta. | The sky turned dark yesterday and it rained. |
| 동생이랑 싸워서 엄마한테 혼났다. | tongsayngilang ssawese emmahanthey honnassta. | My mom scolded me for fighting with my younger sibling. |
| 시원한 물 한 잔 주세요. | siwenhan mwul han can cwuseyyo. | I would like a glass of cold water. |

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
