# Peer review of "Prosody-Based Measures for Automatic Severity Assessment of Dysarthric Speech"

_applsci, doi:10.3390/app10196999_

Round 1

Reviewer 1 Report

The objective of the study entitled "Prosody-Based Measures for Automatic Severity Assessment of Dysarthric Speech" is to provide objective means of classifying dysarthria in terms of severity level by utilizing prosody-based measures. This is an interesting study that fills an important gap in the relevant literature concerning the classification of dysarthria via machine learning. The paper is methodologically sound and well-written, and it will be of great interest to both academics in the field of acquired language disorders, and clinicians in medical settings.

There are only a few minor points I would like to suggest:

(1) The Introduction could also include citations of studies on dysarthria occurrence in primary progressive aphasia, especially, in the nonfluent/agrammatic variant

(2) Since the study focuses on the prosodic properties of English and Korean, the authors should include a separate section in the Introduction that would briefly discuss the prosodic properties of the two language systems

(3) Method. Did the authors take into account the geographic origin of the patients whose speech production was included in the English and Korean corpora? Various aspects of prosody are closely tied with the socio-geographical origin or/and the idiolect of an individual. I think this is an important parameter that should have been taken into consideration by the authors in the analysis of the data.

Reviewer 2 Report

Neurological disorders come with a range of cognitive and physical issues that can make individual’s life difficult. Among them, speech is one aspect that can be severely damaged and lead to issues in communication. In this manuscript, Dr. Minhwa Chung and co-workers presents an automatic method of assessing dysarthria in a range of severity levels using prosody-based measures. Thus they attempted to demonstrate the benefits of prosody for dysarthria severity assessment.

The overall representation of the manuscript is comprehensively described, the study details are described and analyzed appropriately. However, the authors still need to make a minor correction before this article is finally accepted for publication by Appl. Sci..

Particular comments:

  • The reference formatting does not conform to requirements of Sci., including reference format coherence (with or without DOI numbers), volume (Ref. 30), page numbers (Ref. 38) etc. The senior author needs to read the guideline of reference guidance of Appl. Sci. carefully and proofread the whole manuscript with great caution.
